# Carpal Tunnel Syndrome in Elite Female Tug-of-War Athletes: Prevalence and Risk Factor Analysis

**DOI:** 10.3390/diagnostics14192120

**Published:** 2024-09-25

**Authors:** Chiang-Hui Huang, Kuo-Cheng Liu, Ju-Wen Cheng, Shao-Chih Hsu, Chih-Kuang Chen

**Affiliations:** 1Department of Physical Medicine and Rehabilitation, Chang Gung Memorial Hospital at Linkou, Taoyuan 333, Taiwan; mpq200@cgmh.org.tw (C.-H.H.); monhh159@cgmh.org.tw (K.-C.L.); winniesheu@msn.com (S.-C.H.); 2Department of Physical Medicine and Rehabilitation, New Taipei Municipal TuCheng Hospital (Built and Operated by Chang Gung Medical Foundation), TuCheng, New Taipei City 236, Taiwan; 3Master of Science Degree Program in Innovation for Smart Medicine, Chang Gung University, Taoyuan 333, Taiwan; 4Graduate Institute of Management, Chang Gung University, Taoyuan 333, Taiwan; 5School of Medicine, College of Medicine, Chang Gung University, Taoyuan 333, Taiwan; 6Department of Physical Medicine and Rehabilitation, Chang Gung Memorial Hospital at Taoyuan, Taoyuan 333, Taiwan; sophie1976@gmail.com; 7Center of Comprehensive Sports Medicine, Chang Gung Memorial Hospital at Taoyuan, Taoyuan 333, Taiwan

**Keywords:** Tug-of-War (TOW), carpal tunnel syndrome (CTS), carpal tunnel pressure (CTP), athlete

## Abstract

**Background:** Tug-of-War (TOW) games involve repetitive hand movements and vigorous force, raising the risk of peripheral neuropathy in the upper extremities. The prevalence of carpal tunnel syndrome (CTS) in TOW athletes remains unclear. We hypothesize that elite female TOW athletes have a higher prevalence of CTS than the general population. **Methods:** Twenty-nine female TOW athletes were recruited from a national team and participated in the study. CTS was clinically diagnosed by history taking and physical examination. Nerve conduction studies (NCS) were additionally performed to confirm CTS. **Results:** Twelve athletes were clinically diagnosed with CTS; however, only nine were confirmed by NCS. Ten athletes were diagnosed with subclinical CTS by NCS, while seven were classified as truly-non-CTS by both clinical assessment and NCS. The prevalence of CTS and subclinical CTS among the athletes was found to be 33.3% and 37.0%, respectively, significantly higher than 2.7% in the general population by electrodiagnosis. The body weight (*p* = 0.025) of the athletes with CTS and subclinical CTS was significantly different from those of the athletes without CTS. **Conclusions:** Our observations revealed a higher prevalence of CTS among elite female TOW athletes, with body weight being a risk factor. The forceful grasping and pulling of the rope may contribute to the development of CTS.

## 1. Introduction

Tug-of-War (TOW) is a popular non-contact sport, in which athletes grasp and pull a rope against an opposite team with maximal force. According to Smith and Krabak, strain and sprain are the most common related injuries in TOW athletes and occur mainly in the back (42%), shoulder–upper limbs (23%), and knee (17%) [1]. An investigation into athletes participating in the Tug-of-War World Outdoor Championship similarly found that TOW-related injuries were located most commonly in the lower back (44%) and lower extremities (56%) [2]. Other sport-related injuries in TOW athletes include joint and extremity injuries [3,4,5,6,7,8,9], visceral injuries [10,11,12], rib and thoracic cage injuries [13,14], and retinal injuries [15]. However, while many musculoskeletal injuries in TOW athletes have been reported, the literature contains very few studies on nerve injuries. In fact, a literature review found only four related publications [11,12,16,17]. Notably, these publications are case reports, describing acute nerve injuries involving the brachial plexus, T10 spinal cord, ulnar nerve, and radial nerve, respectively. To date, the literature lacks well-designed studies discussing chronic peripheral neuropathies, such as carpal tunnel syndrome (CTS), in TOW athletes.

CTS is an entrapment neuropathy of the median nerve and is the most common form of peripheral neuropathy among the general population [18]. According to Atroshi et al., the prevalence of CTS among the general population is 4.9% by clinical diagnosis and 2.7% by electrodiagnosis [19]. The major risk factors of CTS include female gender [20], obesity, pregnancy, menopause, trauma, and certain medical diseases [21]. However, sport is also a potential risk factor for CTS. For example, in a case series of over 200 injuries, Krivickas found that CTS is the most common peripheral neuropathy among athletes and has a prevalence rate of around 12.4% [22].

TOW games are characterized by forceful grasping and pulling of the rope with maximal effort. Under long-term training, overgrowing callus over the palmar aspect of the wrists can be observed in many elite TOW athletes (Figure 1). Moreover, TOW athletes are exposed to excessive pressure over the carpal tunnel and are thus at high risk of CTS. However, the literature lacks any formal reports on the association between TOW and CTS. Notably, CTS development is often characterized by an absence of specific symptoms or signs due to strong muscle bulk. Consequently, the diagnosis of CTS may be delayed, resulting in progression to more severe nerve injury. Accordingly, this cross-sectional investigation aims to assess three issues, namely, (1) the prevalence of CTS, (2) the prevalence of subclinical CTS, and (3) the risk factors of CTS among elite female TOW athletes. In performing the research, it is hypothesized that CTS is more prevalent in elite female TOW athletes than among the general population.

## 2. Materials and Methods

### 2.1. Participants

Elite female athletes from a national TOW team were recruited in this study. The inclusion criteria were as follows: female, aged between 15 and 25 years old, and training experience of more than one year. The exclusion criteria were athletes who refused to be enrolled, significant trauma history of upper extremities, and previously diagnosed as CTS. The study was performed according to the Declaration of Helsinki and was approved by the Institutional Review Board of Chang Gung Memorial Foundation, Taiwan. All of the subjects were given a provided written informed consent form after the experimental procedures were explained [23].

### 2.2. Study Design

The participants completed a personal history questionnaire, including items such as age, body height, body weight, body mass index (BMI), past medical history, TOW seniority, CTS-specific symptoms, such as numbness or tingling sensation, Boston Carpal Tunnel Syndrome Questionnaire (BCTQ), and TOW position. (Note that the TOW position indicates the usual pulling position of the participant during training and competition (i.e., position 1 to position 8 from the center of the rope to the end point)). The BCTQ is a widely used self-administered scale that assesses self-perceived symptom severity and functional status. It includes the Symptom Severity Scale (SSS) and the Functional Status Scale (FSS) [24]. A physician conducted a thorough review of the patient’s history and ensured the questionnaire was accurately and completely filled out.

Furthermore, another physician performed a physical examination according to the clinical diagnostic criteria of CTS to each participant. Physical examinations included thenar muscle weakness, Phalen test, and Tinel test for both hands of participants. Thenar muscle weakness is evident when the thumb struggles with tasks like buttoning clothes, gripping a phone, or opening jars, or when there is visible atrophy of the thenar muscle. The Phalen test was conducted by placing the wrist in full, unforced flexion for 60 s, with symptom reproduction or exacerbation considered a positive result. The Tinel test involved tapping the median nerve at the wrist crease, with a positive outcome defined as tingling along the median nerve distribution. Adson test and Spurling test were also performed to exclude proximal nerve lesions.

Finally, the participants underwent a nerve conduction study (NCS) of the median nerve to perform electrodiagnosis of CTS by the other physician. Based on the clinical and NCS outcomes, the participants were classified into three groups, namely, (1) the definite CTS group (i.e., those who met both the clinical and the electrodiagnostic criteria of CTS); (2) the truly-non-CTS group (those who met neither the clinical criteria for CTS not the electrodiagnostic criteria); and (3) the subclinical CTS group (those who met the electrodiagnostic criteria for CTS, but not the clinical criteria). All physicians were blinded to the participant’s medical history prior to the assessments, and the results of the physical examinations and NCS were also kept blinded from each other.

Two of the athletes suffered from median nerve neuropathy and were diagnosed with CTS before. Treatment with median nerve hydrodissection was given, resulting in a significant reduction in discomfort. These two athletes were excluded from this study.

### 2.3. Clinical Diagnostic Criteria

The clinical diagnostic criteria for CTS were specified as follows: (1) numbness or a tingling sensation over the digital median nerve-innervated region, or thenar muscle weakness; and (2) a positive Phalen test result or positive Tinel test result. For each participant, CTS was diagnosed if both criteria were met.

### 2.4. Nerve Conduction Studies

All of the participants underwent nerve conduction studies (Nicolet Viking Select, Madison WI, USA) after the clinical diagnostic test. The studies were performed under room temperature conditions (a minimum surface body temperature of 32 °C). The motor and sensory responses of the median nerves of both hands were examined. The motor response was recorded over the abductor pollicis brevis muscle with stimulation at 8 cm proximal to the recording electrode. The sensory response was recorded antidromically over digit 2 with stimulation at two points, namely, 7 cm proximal to the recording electrode in mid-palm and 14 cm proximal to the recording electrode over the wrist [25,26]. For both responses, the onset latency, amplitude from base to peak, and conduction velocity were all recorded.

In the case of equivocal findings of the motor and sensory responses, the Combined Sensory Index (CSI) was further examined. A positive CSI was defined if any of the following criteria were met: (1) median and radial antidromic conduction at 10 cm (significant if >0.5 ms); (2) median and ulnar antidromic conduction at 14 cm (significant if >0.4 ms); (3) median and ulnar orthodromic conduction at 8 cm (significant if >0.3 ms); and (4) total combined index above >0.9 ms [27].

The diagnostic criteria of NCS for CTS included the following: (1) median nerve digit 2-wrist sensory velocity < 50 m/s, (2) median nerve distal motor latency ≥ 4.2 ms, or (3) a positive CSI.

### 2.5. Statistical Analysis and Risk Factor Analysis

Statistical analysis was conducted using PASW Statistics 18 (SPSS Inc., Chicago, IL, USA). The experimental data were expressed as mean ± standard deviation (SD) values for the continuous variables and the number of counts for the categorical variables. The risk factors, including the athletes’ age, body height, body weight, body mass index (BMI), disease history, and seniority, were analyzed by means of independent sample *t*-tests with a statistical significance level of *p*-value < 0.05.

## 3. Results

### 3.1. Patient Flow and Baseline Data

Twenty-seven female TOW athletes were recruited. Ten of the athletes were diagnosed with CTS by clinical assessment. However, only nine of them were confirmed to have CTS by NCS. One athlete presented with CTS-like symptoms, but the NCS finding was negative. The athlete was very sensitive to all stimuli, but her NCS findings were all negative. For further confirmation, ultrasonography of the median nerve was applied. The results revealed a normal median nerve structure and condition (CSA-I < 10.5 mm^2^ and FR-H < 3.3). Consequently, the erroneous clinical assessment of CTS may be the result simply of hypersensitivity.

After excluding this athlete from the CTS group, the prevalence of CTS among the female TOW athletes was found to be 33.3% (9/27). Among the nine athletes in the CTS group, three had bilateral CTS, while six had unilateral CTS.

Seventeen athletes were classified as non-CTS based on clinical assessment. However, 10 of these athletes were later diagnosed as subclinical CTS through NCS findings that met the diagnostic criteria for CTS, despite the absence of clinical symptoms. Among these 10 athletes, 1 had bilateral subclinical CTS and 9 had unilateral subclinical CTS. Of those athletes with unilateral subclinical CTS, six were right-sided subclinical CTS and three were left-sided subclinical CTS. The overall prevalence of subclinical CTS among the female TOW athletes was 37.0% (10/27).

Seven athletes were classified as truly-non-CTS based on both a negative clinical assessment and a negative NCS outcome.

### 3.2. Risk Factor Analysis

In analyzing the risk factors for CTS, the participants were classified simply as CTS/subclinical CTS groups or non-CTS groups (Table 1). The non-CTS group included the truly-non-CTS group, and the athletes were excluded from the CTS group by NCS. The body weight is significantly different. In particular, the average body weight of the CTS group was heavier than that of the non-CTS group (*p* = 0.025).

No statistically significant difference was found in the age (19.0 ± 2.1 years vs. 19.8 ± 1.8 years, *p* = 0.321), body height (161.8 ± 6.7 cm vs. 159.3 ± 7.3 cm, *p* = 0.400), BMI (27.4 ± 3.5 kg/m^2^ vs. 26.2 ± 1.6 kg/m^2^, *p* = 0.217), disease history, seniority (85.4 ± 30.7 months vs. 98.0 ± 24.4 months, *p* = 0.273), BCTQ severity scale (15.0 ± 4.5 scores vs. 15.9 ± 6.5 scores), or functional scale (8.9 ± 1.56 scores vs. 8.6 ± 1.8 scores) of the CTS/subclinical CTS groups and non-CTS groups.

## 4. Discussion

CTS is the most common peripheral neuropathy in the general population [18]. Its prevalence is much higher among athletes in certain sports, such as wheelchair basketball players [28]. This study has focused on female TOW games, in which repetitive movement and vigorous force output patterns may place excessive pressure on the carpal tunnel and hence lead to CTS.

The present results, based on 27 athletes, have shown a prevalence rate of 33.3% for CTS with typical clinical symptoms/signs and positive NCS findings. This rate is much higher than that among the general population, which was 2.7% [19]. Furthermore, the prevalence of subclinical CTS among the female TOW athletes was 37.0%. This relatively high rate of subclinical CTS may result from a higher tolerance of the athletes toward discomfort. In other words, it is possible that the athletes simply ignored the feelings of discomfort associated with CTS. Elite athletes have greater strength than the general population and are less bothered by slight weakness or physical discomfort. In other words, while objective examinations, such as NCS, may disclose abnormal findings, the athletes may exhibit no clinical CTS symptoms or signs. Moreover, the high prevalence of CTS in this age group is inconsistent with the general population’s distribution, where the highest prevalence typically occurs in individuals in their 50s, and the prevalence among those in their teens to 20s is less than 5% [29].

Previous studies had identified obesity as a risk factor for CTS in the general population [21,30]. Our results indicate that body weight is a potential risk factor for developing CTS among elite female TOW athletes. Despite no statistical significance, body height in the CTS and subclinical CTS groups was higher than in the non-CTS group, which diluted the significance of the body weight differences, resulting in a non-significant difference in BMI between the two groups. Since these elite athletes were classified as overweight to obesity status based on the BMI cutoffs in Taiwan [31], increased body weight is a significant risk factor for developing CTS. Besides this, the elevated BMI among these elite female TOW athletes primarily results from the greater muscle mass demand for strength-dependent sports, which is distinct from obesity in the general population. The forceful grasp from the forearm muscles demands increased muscle strength, leading to greater muscle mass, higher body weight, and potentially greater load on the wrist, contributing to CTS. Therefore, for these elite female TOW athletes, body weight (BW) may be a more important risk factor than BMI.

Although there is no statistical significance in seniority, the seniority in CTS and subclinical CTS groups is less than in non-CTS groups. Comparing the athletes from high school and university, training-loading among high school athletes is heavier than university athletes. Furthermore, junior athletes were less skillful than senior athletes. The senior athletes used techniques to protect themselves from sports-related injuries, but the junior athletes were less experienced and more susceptible to sports-related injuries. Therefore, CTS may be more prevalent among younger and more junior athletes.

Neither of the BCTQ scales showed statistical significance. Interestingly, the SSS score was higher in the non-CTS group compared to the CTS and subclinical CTS groups. Upon reviewing the questionnaire, it appears that TOW athletes often experience numbness and pain in the wrist and hand after training. This overlap in symptoms makes it challenging to accurately assess the severity of CTS using the BCTQ and to differentiate between CTS and the effects of training. This finding further confirms that BCTQ appears to lack accuracy in assessing CTS among these elite female TOW athletes.

TOW training and competitions include forceful grasping and pulling of the rope through the upper extremities, which may increase the load on the wrist, raising the possibility of CTS. A previous study found that the wrist and finger muscles in the upper extremities demonstrated greater activation compared to the proximal muscles during pulling and holding the rope [32]. The co-contraction of both extensors and flexors of wrist and finger muscles increases the stiffness of the wrist joint to deliver pulling force to the rope. The higher co-contraction of muscles can lead to higher compressive forces on the joint [33]. Furthermore, increasing carpal tunnel pressure (CTP) leads to decreased microcirculation and injury to the median nerve [34], which is one of the pathophysiologic mechanisms of CTS [35]. Goss, B.C. and J.M. Agee used a percutaneous pressure transducer, which is an invasive device, to measure the CTP [36]. The pressure transducer was inserted into the carpal tunnel and used to evaluate CTP during different grip forces. The results showed that higher grip force could cause increased CTP. Furthermore, McGorry, R.W., et al. also used a percutaneous pressure transducer to evaluate the change in CTP during different grip types, resistance forces, and wrist positions and found that there was an increase in CTP as the resistance force became larger [37]. Therefore, in TOW athletes, the high co-contraction of wrist and finger muscles, along with resisting the contra-directional force of the rope, may be associated with increased CTP, potentially increasing the possibility of CTS.

Studies have shown that being female is a risk factor for developing CTS. Female during pregnancy and after menopause have a higher prevalence [38], suggesting that hormonal changes may contribute to this association. However, in this study, none of the participants were pregnant, had a history of pregnancy, or after menopause. Therefore, the high prevalence in these elite female TOW athletes may relate to their high muscle composition and intense training load, rather than hormonal changes.

Beyond CTS, symptoms and signs of cervical radiculopathy and left ulnar neuropathy were also noted among the TOW athletes. While not as prevalent as CTS, these symptoms and signs may also be related to the force direction and pulling position of the athlete. Further investigation is thus required in future studies.

Several limitations of this study should be noted. Firstly, the included sample size is limited. Although TOW is a popular sport, the number of elite athletes involved is lower than expected. Through this study, we recruited one elite female TOW team to investigate the prevalence of CTS. In other words, a total of just 27 athletes was taken to represent elite female TOW athletes. Therefore, great care should be exercised when applying the current findings to the general female TOW population, as they typically have a lower training load. While the study design used in this study seems valid, future studies should attempt to recruit a greater number of TOW athletes of the same level or with a similar training intensity in order to validate the present findings. Secondly, a comparative study with the general population of the same geographical area and age group could highlight the impact of the sport. Thirdly, the present study was a cross-sectional study and focused only on short-term follow-up. Long-term follow-up studies should be considered in the future. Lastly, the changes in the CTP were not measured in this present study. Due to the invasive nature of the percutaneous pressure transducer, it may not be suitable to assess CTP in elite TOW athletes. In the future, a non-invasive CTP measurement should be designed to investigate the changes in the CTP and the association with the CTP and the CTS among TOW athletes.

## 5. Conclusions

Our observations revealed a higher prevalence of CTS among elite female TOW athletes compared to the general population. Body weight is a risk factor affecting the prevalence of CTS in such athletes with overweight to obese status. Furthermore, the intense training load involving forceful grasping and pulling of the rope in TOW athletes may potentially increase CTP, which could be a risk factor for developing CTS. In general, the results suggest that further investigation is required to examine appropriate CTS treatment and prevention strategies for elite female TOW athletes.

## Figures and Tables

**Figure 1 diagnostics-14-02120-f001:**
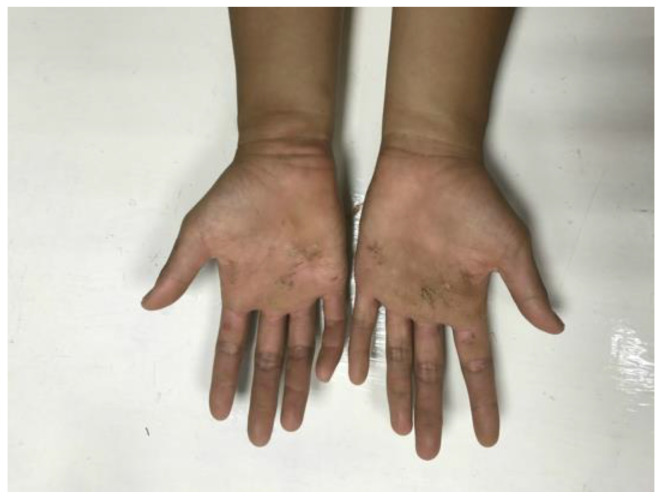
Wrists of TOW athletes. Overgrowth of callus was observed on the palms of the wrists.

**Table 1 diagnostics-14-02120-t001:** Baseline data between CTS/subclinical CTS group and non-CTS group.

	CTS/Subclinical CTS Groups (*n* = 19)	Non-CTS Group (*n* = 8)	*p*-Value
Age (years)	19.0 (±2.1)	19.8 (±1.8)	0.321
Body height (cm)	161.8 (±6.7)	159.3 (±7.0)	0.400
Body weight (kg)	71.4 (±6.2)	66.3 (±4.4)	0.025 *
BMI (kg/m^2^)	27.4(±3.5)	26.2 (±1.6)	0.217
Seniority (months)	85.4 (±30.7)	98.0 (±24.4)	0.273
BCTQ SSS	15.0 (±4.5)	15.9 (±6.5)	0.737
BCTQ FSS	8.9 (±1.56)	8.6 (±1.8)	0.715

CTS: carpal tunnel syndrome; Non-CTS: non-carpal tunnel syndrome. * *p*-value < 0.05.

## Data Availability

Due to privacy concerns, the data presented in the study are available from the corresponding authors upon reasonable request.

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
