# Peer review of "Carpal Tunnel Syndrome in Elite Female Tug-of-War Athletes: Prevalence and Risk Factor Analysis"

_diagnostics, 2024, doi:10.3390/diagnostics14192120_

Round 1

Reviewer 1 Report

Comments and Suggestions for Authors

Interesting and well-written manuscript. Here are a few tips to improve it:

Line 17: as it is first time you use this abbreviation, specify CTS

Lines 45-47: edit English

Line 77: no exclusions criteria? (past traumas, symptoms similar to CTS before beginning TOW…)

Line 81: specify which data was collected

Lines 79-91: specify if the same examiner performed all the clinical examinations, the same thing for the NCS.

Line 95: Phalen test is positive or negative at a specific time… you should precise something like “Phalen test positive within 10 (or 15) seconds.”

Lines 129-132: I believe (according to the comment for line 77) that those two athletes should be excluded from this study to avoid bias.

Lines 163-166 have to be deleted!!

In the discussion, you should also mention something about age… Your patients are between 15 and 25 years old, but CTS in the general population appears surely later.

Line 182-192: try to explain why body weight could be a risk factor and not BMI (it would make more sense).

Lines 214-227: in the limitations, please add biases due to the lack of exclusion criteria… maybe mention that a comparative study with a general population of the same geographical area and same age could be useful

Lines 235-237  have to be deleted!!

Comments on the Quality of English Language

minor editing could be useful (see comments above)

Reviewer 2 Report

Comments and Suggestions for Authors

Thank you for the opportunity to review the manuscript entitled "Carpal Tunnel Syndrome in Elite Female Tug-of-War (TOW) Athletes: Prevalence and Risk Factor Analysis”. The purpose of this study was to assess the prevalence and risk factors for carpal tunnel syndrome in Elite Female Tug-of-War Athletes. Based on the conducted research and analysis of the results, the authors conclude that elite female TOW athletes have a high prevalence of CTS, with body weight being a significant risk factor.

It should be emphasized that the topic of the work is very interesting, and the lack of research confirms the validity of its undertaking.

Unfortunately, it appears that examining only 29 female TOW athletes is not sufficient to assess the incidence of CTS and identify risk factors.

The authors set three goals for this work: : (1) the prevalence of CTS, (2) the prevalence of subclinical CTS, and (3) the risk factors of CTS among female TOW athletes. Is it possible to answer these questions based on examining 29 female TOW athletes? Will this be a reliable and credible study based on 29 female TOW athletes?

There is also an important explanation for many aspects of the methodology of this research!

Line 80-81. What is this questionnaire? What do CTS-specific symptoms mean? This must be described in detail! It is better to use a validated questionnaire, e.g. Boston Carpal Tunnel Questionaire!

Who performed the physical examination? Was the examiner blinded?

Line 94: "..thenar muscle weakness". Who assessed it and how?

Line 130: Previously treated people should be excluded. There are already 27 people!

Line 141: The authors write about subclinical symptoms of CTS based on the NCS study, but they do not specify what these subclinical NCS values ​​are!

Line 146: The authors write "The overall prevalence of subclinical CTS among the female TOW athletes was 34.5% (10/29)." What about the 3 athletes presenting with CTS-like symptoms (Line 127)?

Line 152: Could a 5 kg mean difference be a risk factor for CTS among 29 young women with a statistically insignificant BMI?

The discussion is a very poorly written part of this paper!

It is impossible to draw such conclusions with such a small group of people studied!
